DISCOVERY REPORT

# Unprecedented female mutation bias in the aye-aye, a highly unusual lemur from Madagascar

Richard J. Wang[1], Yadira Peña-García[1], Muthuswamy Raveendran[2,3], R. Alan Harris[2,3], Thuy-Trang Nguyen[4], Marie-Claude Gingras[2,3], Yifan Wu[2], Lesette Perez[2], Anne D. Yoder[5], Joe H. Simmons[6], Jeffrey Rogers[2,3], Matthew W. Hahn [1,4]*

**1** Department of Biology, Indiana University, Bloomington, Indiana, United States of America, **2** Human Genome Sequencing Center, Baylor College of Medicine, Houston, Texas, United States of America, **3** Department of Molecular and Human Genetics, Baylor College of Medicine, Houston, Texas, United States of America, **4** Department of Computer Science, Indiana University, Bloomington, Indiana, United States of America, **5** Department of Biology, Duke University, Durham, North Carolina, United States of America, **6** Keeling Center for Comparative Medicine and Research, MD Anderson Cancer Center, Bastrop, Texas, United States of America

\* mwh@iu.edu

The Editors encourage authors to publish research updates to this article type. Please follow the link in the citation below to view any related articles.

## Abstract

Every mammal studied to date has been found to have a male mutation bias: male parents transmit more de novo mutations to offspring than female parents, contributing increasingly more mutations with age. Although male-biased mutation has been studied for more than 75 years, its causes are still debated. One obstacle to understanding this pattern is its near universality—without variation in mutation bias, it is difficult to find an underlying cause. Here, we present new data on multiple pedigrees from two primate species: aye-ayes (*Daubentonia madagascariensis*), a member of the strepsirrhine primates, and olive baboons (*Papio anubis*). In stark contrast to the pattern found across mammals, we find a much larger effect of maternal age than paternal age on mutation rates in the aye-aye. In addition, older aye-aye mothers transmit substantially more mutations than older fathers. We carry out both computational and experimental validation of our results, contrasting them with results from baboons and other primates using the same methodologies. Further, we analyze a set of DNA repair and replication genes to identify candidate mutations that may be responsible for the change in mutation bias observed in aye-ayes. Our results demonstrate that mutation bias is not an immutable trait, but rather one that can evolve between closely related species. Further work on aye-ayes (and possibly other lemuriform primates) should help to explain the molecular basis for sex-biased mutation.

## Introduction

Male and female mammals transmit de novo mutations (DNMs) at vastly different rates [1–3]. In every mammalian species studied to date, males transmit more mutations than females across their reproductive life span, as well as showing a strong effect of parental age on the number of transmitted mutations from fathers [4–12]. In contrast, it was not clear that there

**Data availability statement:** All relevant mutation data are within the paper and its Supporting information files. All raw sequencing data from aye-ayes and baboons sequenced here are available from the Sequence Read Archive, BioProject numbers PRJNA1156176 and PRJNA1156185, respectively.

**Funding:** Funding for this study was provided by grants from the National Institutes of Health, grant number: R01-HD107120 awarded to M.W.H. and P40-OD024628 to J.H.S. The funders played no role in the study design, analysis, decision to publish, or preparation of the manuscript.

**Competing interests:** The authors have declared that no competing interests exist.

**Abbreviations :** DNMs, de novo mutations; DLC, Duke Lemur Center; GQ, genotype quality; HAL, hierarchial alignment format; MAF, multiple alignment format; SNPs, single-nucleotide polymorphisms; SNVs, single-nucleotide variants; VCF, variant call format.

was a significant parental age effect among mothers until whole-genome mutation studies were conducted on hundreds of trios in humans [6,13,14].

Male mutation bias has long been attributed to the persistence of replication in the male germline post-puberty, a phenomenon absent from the female germline [15]. This classic and straightforward hypothesis has, however, become challenged as many pieces of evidence now suggest that germline replication may not be the major determinant of male mutation bias [16,17]. One obstacle to understanding the underlying causes of male mutation bias is its ubiquity across studied species. Indeed, fish are the most closely related organism to mammals without an apparent male bias [1]. Not only does the male parent consistently transmit more mutations to its offspring across mammals, the rate at which these mutations accumulate with paternal age appears to be highly similar across mammals [4,5,8,10,11]. Without variation in the sex-specific pattern for these traits, it is difficult to further investigate their causes.

The strepsirrhine primates—a clade including lemurs, lorises, and galagos—comprise about half of living primate species, but are relatively unstudied with genomic approaches. One important reason for the lack of genomic studies in this group is that most of the species are endangered [18], with all lemuriform primates found only on the island of Madagascar where they are severely threatened by anthropogenic habitat destruction and climate change. The single study of mutation rates in this group (in gray mouse lemurs, *Microcebus murinus*) showed the lowest male mutation bias of any mammal reported to date, though this estimate is underpowered, coming from only two trios [19]. Given that *M. murinus* is only one of nearly 100 lemuriform species, there is an obvious need to further characterize mutation rates in the clade before drawing any firm conclusions about their transmission characteristics.

The aye-aye (*Daubentonia madagascariensis*) is a nocturnal lemur known for a number of distinctive characteristics [20]. Most notably, this species has evolved a unique feeding strategy, using a long and thin third digit to tap on branches to "echolocate" for burrowing insects and other prey. The aye-ayes dig their prey out by gnawing through the woody substrate with their rodent-like ever-growing incisors. They then capture their prey using this elongated middle finger, which is supported by a ball-and-socket joint allowing 360° of movement. These traits are unique among primates [21,22]. Aye-ayes also have a long life span relative to their body size, in some cases living more than 30 years and reproducing at much older ages than other lemuriforms [23]. Their distinctive morphology and behavior have evolved through an extensive period of evolutionary independence: *D. madagascariensis* is the only extant member of the Daubentoniidae primate family and is estimated to have diverged from all other Malagasy lemurs at least 50 million years ago [24,25].

In this study, we report the sequencing of 18 aye-aye whole genomes from a large pedigree consisting of 12 trios (Fig 1A). Additionally, we sequence 9 olive baboons (*Papio anubis*) comprising 4 trios (Fig 1B), which we combine with 25 previously sequenced individuals [12]. Applying a consistent set of methods to identify DNMs and to assign them to a parent of origin (i.e., "phase" them), we find that in aye-ayes maternal age has a much stronger effect on the number of transmitted mutations than paternal age. We compare aye-ayes to data from baboons, rhesus macaques, and humans to show that this finding is not a function of differences in sample preparation or computational methods. We then search a curated set of DNA replication and repair genes for amino acid substitutions either unique to aye-ayes or shared among all available lemuriform primate genomes to identify possible candidate mutations leading to this unprecedented female bias. The results presented here point to a new approach for uncovering and understanding the causes of sex-biased mutation.

## A  Aye-ayes

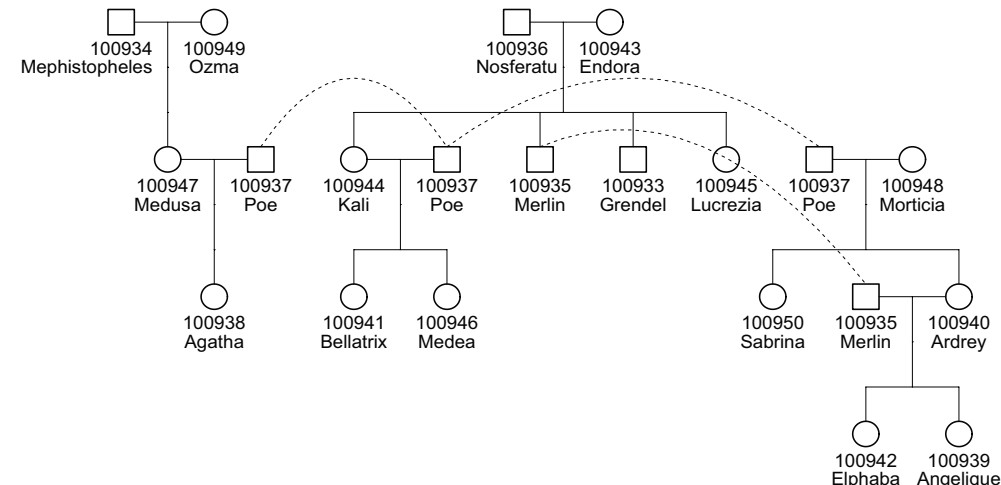

## B  Baboons

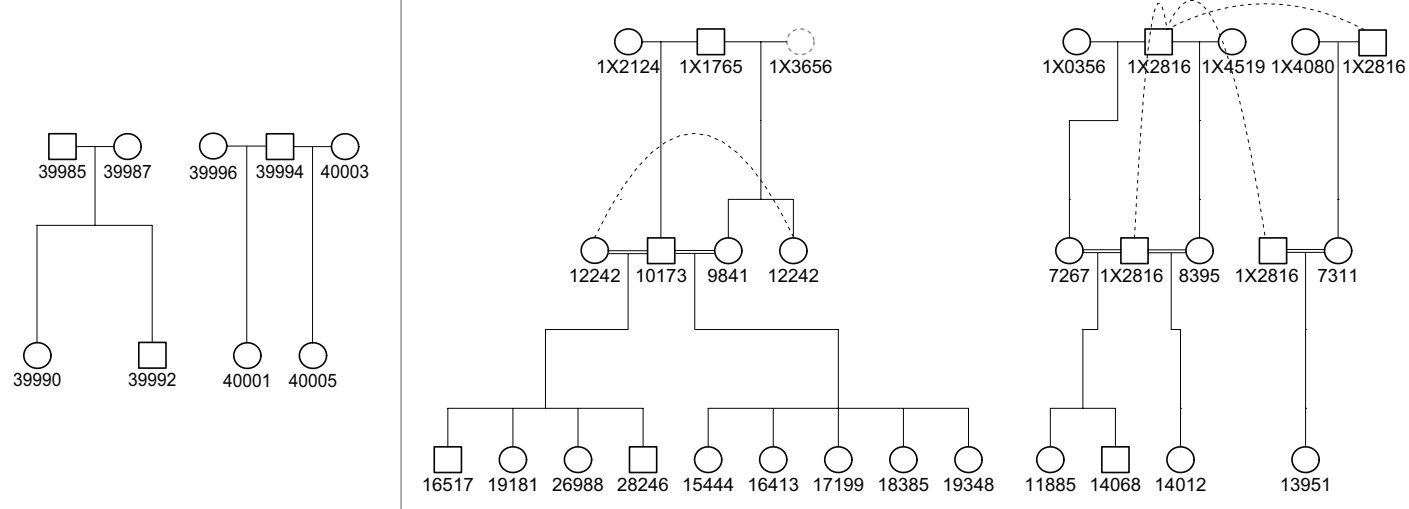

**Fig 1. Primates used in this study.** (**A**) Pedigree structure of aye-ayes. The 18 individuals sequenced in this study are shown: males as squares and females as circles. Two individuals (IDs 100937 and 100935) appear multiple places in the pedigree, each time connected by a dashed line. Offspring IDs in Table 1 refer to individuals here. (**B**) Pedigree structure of baboons. The nine individuals sequenced in this study are shown on the left hand side. The 25 individuals sequenced in Wu and colleagues [12] are shown on the right hand side. One individual (ID 1X3656; dashed gray circle) had low read-mapping and was not used here (and was not counted among the 25 individuals). Consanguineous matings are represented as double-horizontal lines, and individuals appearing in multiple places are again connected by dashed lines. Offspring IDs in Table 2 refer to individuals here.

## Results

### Identifying de novo mutations in two primates

**Mutations in aye-ayes.** We sampled 18 individuals from an extended pedigree of aye-ayes housed at the Duke Lemur Center (DLC) in Durham, North Carolina (Fig 1A). These individuals can be divided into 12 independent trios (mother, father, and offspring), as several pairs of parents had multiple offspring together and several individuals had offspring with multiple different partners. After sequencing with Illumina short-read technology (Materials and methods), we obtained an average of 39.1X whole-genome coverage. The sex for each

individual assigned by the DLC was confirmed using read-depth and heterozygosity on X-linked scaffolds (S1 Fig).

We used a set of computational methods and candidate filters shown to have high accuracy [26] to find autosomal DNMs in each pedigree (Materials and methods). In total, we identified 647 DNMs across the 12 trios (Tables 1 and S1). While we only attempted to call mutations consisting of single-nucleotide variants (SNVs), we found one multinucleotide mutation generated by two neighboring SNVs in individual 100940 (cf. [27]). Two mutations (not near each other) were also shared between the same two siblings, individuals 100933 and 100944 (S1 Table). All these mutations were included in our final count. For five of the trios, we were also able to assess the fraction of identified mutations that were transmitted to the next generation. In those trios, we found 160 out of 327 (49%) mutations transmitted, which is not significantly different from the 50% expectation for true germline variants ($P = 0.74$, Exact test). Accounting for the total number of sites at which DNMs could be identified ("callable genome size" in Table 1), we calculate an aye-aye mutation rate of $1.49 \times 10^{-8}$ per bp per generation for parents that conceived at an average age of 15.4 years across both sexes. This average mutation rate is among the highest observed in mammals [1,26], consistent with the advanced average age of the parents sampled here and the relatively high mutation rate observed in the one other strepsirrhine primate for which the DNM rate has been measured [19].

The mutation spectrum in aye-ayes is similar to that found in other non-strepsirrhine primates (S2A Fig), with a transition to transversion ratio (Ti:Tv) of 2.59 and a large fraction of mutations at CpG sites (17.7% of all mutations). However, these results are not consistent with the low Ti:Tv ratio (0.96) and smaller fraction of CpG mutations (8.7%) observed in the gray mouse lemur [19].

**Mutations in baboons.** We sampled nine individuals from two families of baboons housed at the Keeling Center for Comparative Medicine and Research in Bastrop, Texas (Fig 1B). Both families contain two offspring that are siblings, resulting in four trios from which independent estimates of the mutation rate can be made. Sequencing with Illumina short-read technology (Materials and methods) resulted in an average of 40.9X whole-genome coverage. Additionally, we used Illumina data from 25 olive baboons (in 17 trios) from a previous study [12]; note that 5 of these trios were sequenced by Wu and colleagues [12] but not included in their main results.

**Table 1. Aye-aye mutations.**

| Offspring ID | Parental age at conception (years) | | Mutations | Phased mutations | Mutations of paternal origin | Mutations of maternal origin | Callable genome size ($\times 10^9$ bp) | Mutation rate ($\times 10^{-8}$/bp) |
|---|---|---|---|---|---|---|---|---|
| | Paternal | Maternal | | | | | | |
| 100947 | 22.0 | 18.0 | 117 | 30 | 7 | 23 | 1.67 | 3.50 |
| 100938 | 30.5 | 13.7 | 48 | 10 | 9 | 1 | 1.78 | 1.35 |
| 100950 | 16.8 | 14.8 | 25 | 12 | 9 | 3 | 1.86 | 0.67 |
| 100940 | 9.3 | 7.4 | 30 | 12 | 7 | 5 | 1.86 | 0.81 |
| 100942 | 17.5 | 15.6 | 30 | 11 | 11 | 0 | 1.85 | 0.81 |
| 100939 | 11.3 | 9.4 | 25 | 10 | 9 | 1 | 1.87 | 0.67 |
| 100935 | 8.5 | 10.5 | 44 | 15 | 9 | 6 | 1.86 | 1.19 |
| 100933 | 24.4 | 26.5 | 108 | 39 | 16 | 23 | 1.86 | 2.90 |
| 100945 | 15.6 | 17.7 | 83 | 27 | 11 | 16 | 1.87 | 2.22 |
| 100944 | 12.1 | 14.1 | 75 | 18 | 10 | 8 | 1.83 | 2.05 |
| 100941 | 20.7 | 9.6 | 36 | 12 | 11 | 1 | 1.81 | 1.0 |
| 100946 | 17.7 | 6.7 | 26 | 7 | 4 | 3 | 1.87 | 7.0 |

Using the same approaches as above (Materials and methods), we identified 519 DNMs across all 21 baboon trios (Tables 2 and S2). Three of these were multinucleotide mutations (within 10 bp of one another) and one (single nucleotide) mutation was shared between siblings; all multi-nucleotide mutations were visually confirmed in IGV [28] and all were counted as mutations. Accounting for the callable genome size (Table 2), we calculate a baboon mutation rate of $0.74 \times 10^{-8}$ per bp per generation for parents that conceived at an average age of 10.7 years across both sexes. The mutation rate we estimate is slightly higher than that previously reported ($0.57 \times 10^{-8}$ per bp; [12]), likely due to the inclusion of males with a higher average age (13.5 years) relative to the males in the previously reported sample (10.7 years in Wu and colleagues [12]).

## Sex-biased mutation in primates

**Female-biased mutation in aye-ayes.** Single-nucleotide polymorphisms (SNPs) near DNMs can be used to help assign the parent-of-origin of DNMs, a procedure sometimes referred to as "phasing." We used the software POOHA [4,29] to identify which parent transmitted the DNMs identified in aye-ayes (Materials and methods). We were able to unambiguously assign 203/647 (31.4%) of aye-aye mutations to one parent or the other (Tables 1 and S1).

Fig 2A shows the number of paternal and maternal mutations as a function of parental age, illustrating a significant association between maternal age and mutation rate (Poisson

**Table 2. Baboon mutations.**

| Offspring ID | Parental age at conception (years) | | Mutations | Phased Mutations | Mutations of paternal origin | Mutations of maternal origin | Mean Depth | Callable genome size ($\times 10^9$ bp) | Mutation rate ($\times 10^{-8}$/bp) |
|---|---|---|---|---|---|---|---|---|---|
| | Paternal | Maternal | | | | | | | |
| 39990[a] | 5.5 | 6.1 | 29 | 16 | 11 | 5 | 41.3 | 2.01 | 0.722 |
| 39992[a] | 14.7 | 15.4 | 22 | 14 | 13 | 1 | 41.7 | 1.99 | 0.552 |
| 40001[a] | 9.9 | 6.5 | 32 | 19 | 15 | 4 | 39.6 | 1.98 | 0.806 |
| 40005[a] | 7.0 | 6.9 | 18 | 12 | 8 | 4 | 40.8 | 2.03 | 0.443 |
| 16517 | 9.5 | 6.0 | 18 | 12 | 10 | 2 | 45.0 | 1.62 | 0.556 |
| 19181 | 12.2 | 8.7 | 20 | 8 | 3 | 5 | 49.9 | 1.48 | 0.675 |
| 26988 | 14.5 | 10.9 | 31 | 13 | 12 | 1 | 47.6 | 1.53 | 1.011 |
| 28246 | 15.5 | 11.9 | 29 | 16 | 13 | 3 | 45.3 | 1.64 | 0.885 |
| 15444 | 8.2 | 8.7 | 23 | 11 | 10 | 1 | 48.2 | 1.70 | 0.675 |
| 16413 | 9.4 | 9.9 | 13 | 7 | 6 | 1 | 48.4 | 1.71 | 0.379 |
| 17199 | 10.4 | 11.0 | 24 | 10 | 10 | 0 | 49.0 | 1.69 | 0.712 |
| 18385 | 11.3 | 11.8 | 26 | 17 | 14 | 3 | 53.6 | 1.64 | 0.794 |
| 19348 | 12.5 | 13.0 | 11 | 5 | 5 | 0 | 50.2 | 1.72 | 0.319 |
| 7267 | 7.9 | – | 35 | 19 | 10 | 9 | 48.3 | 1.67 | 1.051 |
| 8395 | 9.5 | – | 17 | 11 | 9 | 2 | 30.7 | 1.41 | 0.604 |
| 7311 | 8.0 | 5.3 | 18 | 10 | 7 | 3 | 30.4 | 1.49 | 0.602 |
| 10173[b] | – | 12.9 | 23 | 16 | 15 | 1 | 41.8 | 1.50 | 0.767 |
| 11885[b] | 15.2 | 7.2 | 33 | 10 | 7 | 3 | 55.9 | 1.65 | 0.997 |
| 14068[b] | 18.6 | 10.7 | 39 | 13 | 12 | 1 | 44.8 | 2.00 | 0.976 |
| 14012[b] | 18.5 | 9.0 | 24 | 7 | 6 | 1 | 30.9 | 1.56 | 0.768 |
| 13951[b] | 18.4 | 10.4 | 34 | 10 | 9 | 1 | 29.1 | 1.29 | 1.323 |

[a]Samples newly sequenced here.

[b]Samples sequenced in Wu and colleagues [12] but not included in analyses.

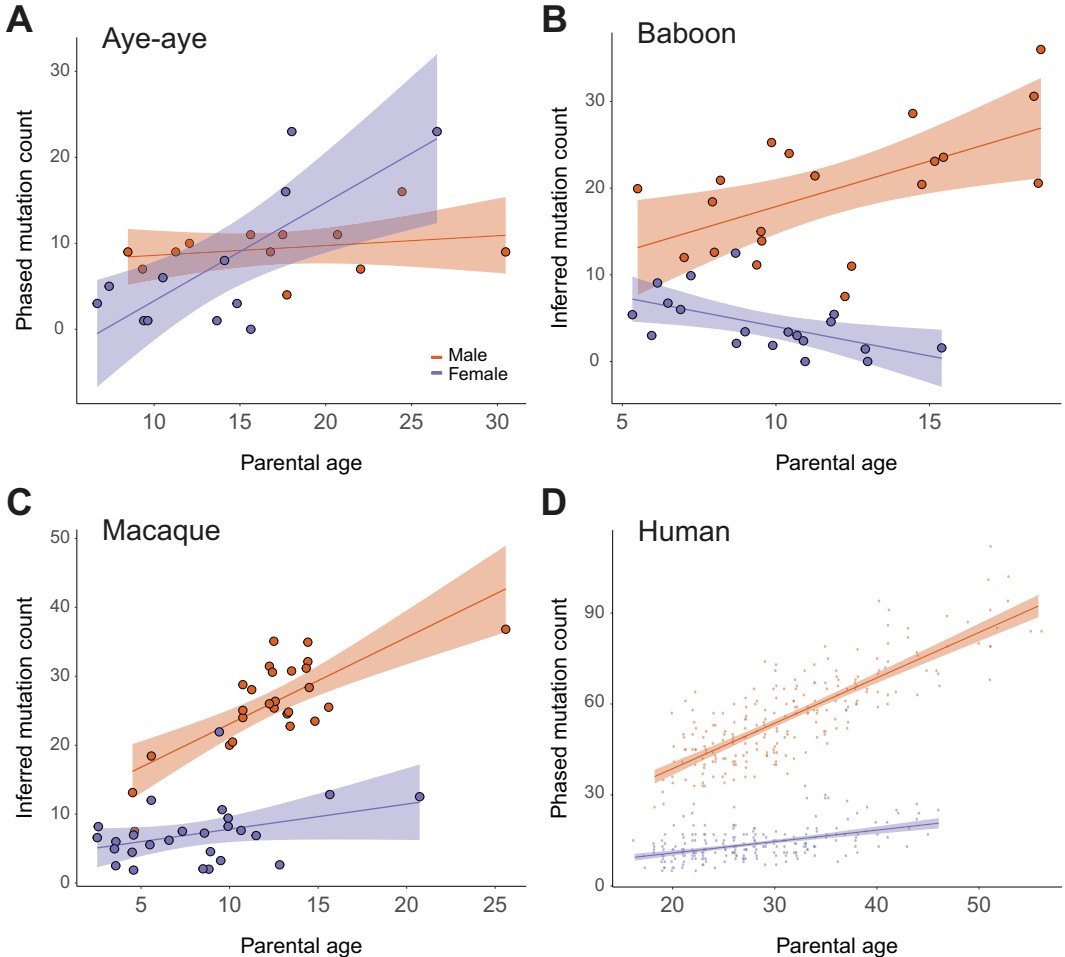

**Fig 2. Relationship between parental age and number of transmitted mutations.** (A) The number of mutations from 12 aye-aye trios assigned ("phased") to either the female (blue) or male (red-orange) parent using the software package POOHA. The *x*-axis shows the age of each parent at conception, while the *y*-axis shows the number of inferred mutations phased for each parent. We performed a Poisson regression on the observed number of phased maternal and paternal mutations and scaled the prediction by the fraction of mutations phased to obtain the regression lines (Materials and methods). Shaded areas show 95% confidence intervals for the regression lines. (B) The number of mutations from 21 baboon trios assigned to each parent. (C) The number of mutations from 26 rhesus macaque trios assigned to each parent. Data come from Wang and colleagues [11] and Bergeron and colleagues [4]. (D) The number of mutations from 225 human trios assigned to each parent. Data come from Jónsson and colleagues [6]. The data underlying this figure can be found in S1 Data.

regression, *P* = 0.03). In contrast to this unexpected relationship, we found no significant association (Poisson regression, *P* = 0.19) between paternal age and the number of paternal phased mutations (Fig 2A). The overall level of sex bias, $\alpha$ = 1.26, is much lower than that observed in other mammals [1,2], with the exception of the gray mouse lemur [19]. The regression analyses reported above used only phased mutations and the percentage of mutations phased (Materials and methods; S3A Fig), but we found a similar association with parental age when the analysis was performed on all mutations, that is, including the mutations that were not phased (Poisson regression, maternal age: $P < 1 \times 10^{-10}$, paternal age: *P* = 0.92; S4 Fig). A recent study of aye-aye mutation rates also found a strong parental age effect on the total number of mutations [30], though these authors were unable to assign most DNMs as coming through the maternal or paternal parent. In our study, we find both that maternal mutations display a strong association

with age and that many more mutations were transmitted by older mothers than older fathers, a pattern not reported for any other mammal and not tested by Versoza and colleagues [30]. We observed no differences among the spectra of mutations transmitted by fathers, older mothers, or younger mothers (S2B–S2D Fig; all $P > 0.3$). Overall, our model predicts an additional 7.2 mutations (95% CI: 5.8, 8.6) in offspring for each additional year of maternal age in the aye-aye.

Previous studies [6,13] have found an enrichment of C > T mutations transmitted by mothers as they age. These excess mutations are likely driven by APOBEC-mediated mutagenesis, which occurs in a TpC dinucleotide context [31]. To determine whether such an effect is present in aye-ayes, we first compared the proportion of TpC > TpT mutations between male and female parents. We identified nine TpC > TpT mutations transmitted by a female parent and five TpC > TpT mutations transmitted by a male parent, which is not significantly different compared to all C > T mutations that could be assigned to a parent (35 paternal and 33 maternal, excluding those occurring at CpG sites; Fisher's exact test: OR = 1.91, $P = 0.38$). To further ask whether there was an effect of maternal age, we compared the proportion of TpC > TpT mutations between young (<15 years) and old (>15 years) mothers. Of the nine TpC > TpT mutations assigned to a female parent, eight were in old mothers and one in a young mother. This distribution is again not significantly different compared to all C > T mutations (28 in old mothers, 5 in young mothers; Fisher's exact test: OR = 1.43, $P = 1.0$). Importantly, while we may not have the statistical power to detect a subtle effect with maternal age, our results do not support an important role for this mechanism in female mutation bias in aye-ayes.

We carried out several analyses to ensure the robustness of the observed patterns in aye-ayes. First, we repeated our entire analysis with an alternative (older and less complete) version of the aye-aye reference genome (DauMad_v1_BIUU) and found nearly identical results: 575/594 mutations found in the alternative reference matched our initial mutation calls. Second, we repeated the phasing of DNMs using the software Unfazed [32], which implements a different algorithm for assigning parent of origin. Unfazed was concordant with POOHA for 196/203 phased mutations. The seven discordant mutations represent cases in which Unfazed missed a call made by POOHA (two paternal and five maternal). Unfazed also phased three mutations that were missed by POOHA, all of which were paternal. Nevertheless, repeating our analyses with only the mutations phased by Unfazed, we found essentially the same qualitative and quantitative results (S3B Fig).

We performed Sanger sequencing to validate all 23 mutations predicted to have been transmitted by the oldest female parent in our study to her last offspring. For these mutations, we sequenced the mother and proband (individuals 100943 and 100933, respectively) as well as the father (100936) and all siblings (100935, 100944, 100945). Sequencing the proband ensures that the mutations are not sequencing errors, while sequencing the parents and siblings ensures that the candidate DNMs were not already present as SNPs in the parents. We confirmed the expected genotypes in all 23 mutations from their presence in the proband and absence in all other individuals (see, e.g., S5 Fig). For four mutations, we were not able to sequence both strands in all individuals, though no contradictory genotypes were observed. Overall, these results indicate the general accuracy of both our mutation calls and the higher number of mutations found in older females.

It is notable that the female bias we observe in aye-ayes is driven by three offspring (100947, 100933, and 100945) arising from only two different mothers (100949 and 100943). Indeed, if we ignore these three offspring, the paternal contribution to DNMs is 74% (equivalent to $\alpha = 2.82$), within the range of 70%–80% reported for most mammals [1,2]. Although these are the oldest ages at conception of any mothers in our dataset, we looked for any further technical differences between these two mothers and other individuals that might

explain the patterns we observe. We verified that all relationships between these mothers and their offspring were correct; further, we checked whether these two mothers were related: they appear to be no more closely related than any other pair of parental individuals in our pedigree (S6 Fig). We observed no significant differences in either read-depth or heterozygosity among individuals, including in the mothers, fathers, and offspring of the most-biased trios (S2 Table). In addition, our set of DNMs were filtered to have an allelic balance >0.35 (S7 Fig), which should eliminate the vast majority of possible somatic mutations (and any that remain should not be assigned to the maternally inherited chromosomes in a biased manner). Finally, as mentioned above, we examined the transmission of mutations detected to the next generation. Overall, this fraction was not different from the expectation of 50%, nor was it different in individual 100947, who transmitted 53/117 detected mutations to their offspring ($P$ = 0.36, Exact test). This result indicates that these mutations are present in the germline of this individual, and that the overall set of detected mutations is unlikely to be composed of somatic mutations.

**Male-biased mutation in baboons, rhesus macaques, and humans.** We repeated the phasing analysis with POOHA on the baboons sequenced in our study and those from Wu and colleagues [12], enabling us to assign 49.3% (256/519) of mutations as being transmitted by one parent or the other (Tables 2 and S3). The higher level of overall nucleotide polymorphism in baboons gave us more power to phase mutations in this species compared to aye-ayes, which have a low level of polymorphism even for strepsirrhines [25,33]. Based on the percentage of mutations that could be phased in each trio, we inferred the number of mutations that each parent transmitted to their offspring.

Fig 2B shows a positive but not statistically significant relationship between paternal age and the number of mutations transmitted by fathers in baboons (Poisson regression, $P$ = 0.20; Materials and methods). However, when the regression analysis is performed on all mutations, we recover a significant association with both paternal and maternal age (Poisson regression, paternal: $P = 1.1 \times 10^{-5}$, maternal: $P = 5.2 \times 10^{-3}$). In contrast to the results from aye-ayes above, and regardless of the effect from paternal age, male parents always transmit more mutations than female parents in baboons (Fig 2B). We note, however, that we did find a negative association between maternal age and the number of mutations (Poisson regression, $P$ = 0.04), which is different than the relationship reported in Wu and colleagues [12]. One possibility for this difference is that a single outlier (ID 18385) drove the positive trend among mothers in Wu and colleagues [12], while we did not find that this individual inherited an unusual number of maternally transmitted mutations (Table 2). A larger number of individuals may be needed to clarify relationships among baboon mothers. To emphasize how different the aye-aye results are from previous findings among primates, in Fig 2C and 2D we show the same plot of phased mutations from rhesus macaques [4,11] and humans [6], respectively (Materials and methods). All three non-strepsirrhine primates show the expected male bias and paternal age effect, in contrast to aye-ayes.

**Estimating sex-biased substitution from comparative data.** We considered whether the bias we observe in our pedigrees has had a long-term effect on nucleotide substitutions by estimating the male-to-female substitution rate ratio, $\alpha$ [34]. This approach compares substitutions on the X chromosome and autosomes across a phylogenetic tree. Sex chromosomes and autosomes are differentially exposed to each of the sexes: over the course of many generations, we expect the X chromosome to spend twice as much time in females compared to males. In contrast, we expect the autosomes to spend an equal amount of time in each of the sexes. Therefore, in the presence of a male mutation bias at the time of reproduction, and assuming the same average age at reproduction, autosomes are expected to accumulate substitutions at a faster rate than the X chromosome, resulting in an $\alpha$ value

greater than 1. Consistent with this expectation, $\alpha$ has been observed to be greater than 1 in all mammals to date [2]. A female mutation bias at the time of reproduction would be expected to generate an $\alpha$ value less than 1.

Our analysis focused on 10 strepsirrhine species, including the aye-aye, to capture a range of evolutionary and ecological contexts (Materials and methods). The estimated $\alpha$ values for these species range from 2.3 on the terminal branch leading to the galago (*Otolemur garnetti*), to as high as 12.3 on the branch leading to the gray mouse lemur (Fig 3). Our estimate for the aye-aye branch was $\alpha = 3.1$, which closely matches the value reported for aye-ayes by de

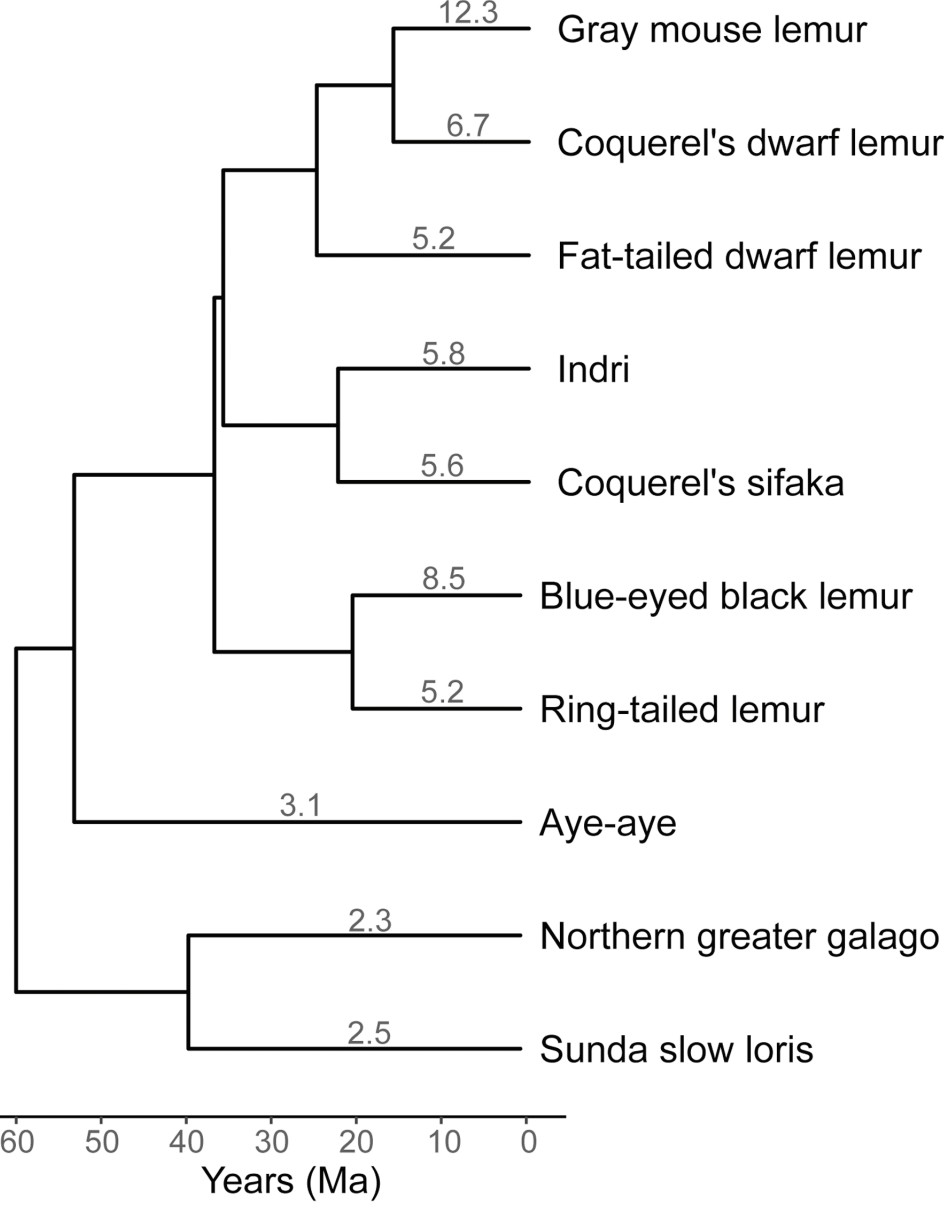

**Fig 3. Sex-biased mutation estimated from phylogenetic comparisons.** The degree of sex-biased mutation over the long-term was calculated by comparing substitutions on the X chromosome to the autosomes (Materials and methods). Values reported for the tip branches of the tree all show $\alpha > 1$, indicating a male-bias at the time of reproduction. Divergence times for the tree come from TimeTree [68]. The data underlying this figure can be found in S1 Data.

Manuel and colleagues [2] ($\alpha$ = 2.95). All branches in the strepsirrhine tree show $\alpha > 1$, indicating a consistent male mutation bias at the time of reproduction.

These results reveal a discrepancy between phylogenetic estimates of sex bias and the one estimated from aye-aye pedigrees. Phylogenetic substitution patterns accrue over millions of years, averaging across the age of reproduction of many different individuals during this time. Therefore, if the average age at which aye-ayes reproduced over the past 50 million years is less than ~15 years (Fig 2A), then the major mutation pattern will likely be one of male bias. To get further insight into long-term patterns of sex-bias, we predicted the age of aye-aye parents that would produce the equivalent of the phylogenetic estimate of $\alpha$ = 3.1, given our pedigree results (Fig 2A). Assuming male and female parents are the same age at conception—and that sex-bias shows a linear relationship with age—aye-aye parents at an average age of 8.97 years would produce the observed phylogenetic level of male-bias. If we instead assume that the three oldest females sampled here are outliers that can be ignored (for instance, because females do not have offspring at these ages in the wild), then $\alpha$ = 2.82, which is within the confidence interval of our phylogenetic estimate (3.1 ± 0.41).

We consider additional explanations that can reconcile the phylogenetic estimates with the pedigree data on mutation rates in the Discussion.

## Searching for possible causative mutations

Regardless of the long-term effects of female mutation bias in aye-ayes, the observed pattern of a stronger effect of maternal age begs a possible mechanistic explanation. To this end, we investigated a list of 219 genes with known roles in DNA replication and repair (S4 Table and Materials and methods), looking for amino acid changes in any of these genes predicted to have deleterious effects by CADD [35]. We carried out three complementary analyses of these data. First, we asked whether the two aye-aye mothers that transmitted higher numbers of mutations (100949 and 100943) shared alleles in any of these genes that were absent from either all other mothers or all other parents. Second, we asked whether there are amino acid substitutions in any of these genes specific to the aye-aye lineage (Materials and methods). Third, given the previous observation of little male-bias in the gray mouse lemur [19], we also looked for damaging amino acid substitutions shared among lemuriforms in any of the 219 genes in our curated list.

We found polymorphisms in and around four genes for which the two aye-aye mothers that transmitted higher numbers of mutations shared an allele not found in all other mothers (S5A Table): these two individuals were heterozygotes, while all other individuals were homozygous for the reference allele. Two of these variants were amino acid changes, and one in the gene *BLM* has a PHRED-scaled CADD score suggesting a deleterious effect. Comparing these two individuals to all other parents (male or female), the amino acid SNP in *BLM* was again only carried by these mothers (S5B Table). The non-synonymous SNP in *BLM* appears to be part of a longer haplotype that is uniquely carried by these mothers. *BLM* encodes a helicase that plays a central role in recombination and meiosis, but the common diseases and cancers that are associated with *BLM* mutations all display recessive inheritance (https://www.ncbi.nlm.nih.gov/gene/641).

Our scan for possibly deleterious substitutions unique to aye-ayes—to the exclusion of all other primates—uncovered many possible candidates (S6 Table). Most strikingly, the gene *PRKDC*, which is involved in DNA double-strand break repair, had five such changes among the top 10 most deleterious substitutions. It also has many other deleterious substitutions shared among aye-ayes and other lemuriforms to the exclusion of other primates (S7 Table). These results suggest that *PRKDC* is no longer functional in the lemuriforms; further manual examination of this gene using the TOGA tool [36] confirmed that it is almost certainly a pseudogene.

The gene *BRIP1* has an amino acid substitution specific to aye-ayes that is predicted to have the most deleterious effect among all detected changes (based on PHRED-scaled CADD scores; S6 Table). *BRIP1* interacts with *BRCA1* and has been implicated in breast, ovarian, and fallopian tube cancers (https://www.ncbi.nlm.nih.gov/gene/83990). Other notable genes with deleterious amino acid substitutions include *MBD4*, which can cause maternal hypermutation [37], and *MUTYH*, in which natural mutator alleles have been found [38] and that may be female-biased [39].

## Discussion

Previous pedigree studies of mutation rates have revealed consistent patterns of male bias across amniotes [1]. Here, we have found two patterns of female bias in a strepsirrhine primate, the aye-aye, that stand in stark contrast to earlier work (Fig 2A). First, aye-ayes show a strong association between maternal age and the number of transmitted mutations, but no association between paternal age and mutations. All mammals studied to date demonstrate the opposite pattern [4–12]. Second, at advanced ages female aye-ayes transmit more mutations to their offspring than do males. Previous studies in mammals have revealed a consistent but contrasting pattern, with males leaving 75%–80% of all mutations across parental ages [4–12,40–44].

We considered multiple factors that could confound the results observed in aye-ayes. Applying the same computational and experimental approaches to baboons revealed the expected patterns of male bias (Fig 2B), as did our own and others' previous work in rhesus macaques (Fig 2C; [4,11]) and published work in humans (Fig 2D; [6]). The baboon results presented here include previously published pedigrees and differ somewhat from the conclusions in that paper [12]. Although the previous study did find the expected excess of male mutation across all ages (82% of all mutations were paternal), it did not find an association between paternal age and the number of transmitted mutations. However, we have added four additional trios that we sequenced, as well as five trios sequenced in Wu and colleagues [12] but not included in their comparison of sex and age. We did not observe any problems with these five trios, and they did not show unusual patterns of mutation (Table 2). Including these additional trios, as well as re-calling and re-phasing DNMs in the original trios, revealed a positive effect of paternal age on the number of mutations, as well as a slightly negative effect of maternal age (Figs 2B and S8). We conclude that the previous study did not observe a correlation between paternal age and mutation rate because a smaller number of trios resulted in imprecise estimates.

Our results in aye-ayes have some precedent in similar results from the gray mouse lemur, another lemuriform primate [19]. This previous study, which included only two offspring that were born less than a year apart from a single pair of parents, also showed a very low fraction of mutations from the male parent (51%) and a relatively high mutation rate. Those results are especially intriguing given that the female parent was 1 and 1.9 years old at conception of the two offspring—sampling an older female might have revealed an even stronger female bias. However, there were also differences between the gray mouse lemur results and aye-ayes: the mouse lemur study showed an unusually low Ti:Tv ratio and an unexpectedly low mutation rate at CpG sites. Both measures were similar between the aye-aye and non-strepsirrhine primates. Given that the mouse lemur DNA was sequenced using a different library preparation method than has been used in other pedigree studies (10x Genomics), how much of the similarity or difference between these studies is biological versus methodological remains to be seen. Regardless, our results provide strong justification for similar studies of pedigree mutation rates in other strepsirrhine primates. It is conceivable that the observations for aye-ayes

may be replicated in other Malagasy lemurs, or even the more distantly related strepsirrhines, the Lorisiformes.

It should be noted that the female-biased results from aye-ayes are driven by three offspring from two different mothers, though these are also the oldest maternal ages in our dataset (Figs 1A and 2A). We looked for technical biases that might have driven the results, but found no differences in either these mothers or their offspring; the mothers are also not closely related (S6 Fig). One of the mothers (individual 100943) has four offspring in the dataset, and it is only the latter two children—born after she was 17 years old—that show an increased number of maternally inherited mutations. This observation argues against the presence of a simple hypermutator phenotype in this individual, though it does not exclude a late-acting phenotype that only appears at an advanced age. It is also true that our aye-aye mothers have had offspring at older ages than any of our baboon samples. However, older rhesus macaque and human mothers have been sequenced, and there is no sign of female-biased mutation at advanced maternal ages in either of these species. Finally, we also observed no effect of paternal age on the number of mutations in aye-ayes. While smaller sample sizes might explain such an observation, when coupled with the results from females it suggests an overall difference in aye-aye reproductive biology.

Given the unprecedented patterns of female bias in aye-ayes, what is the underlying basis for the difference in germline mutation rates? We searched a list of genes known to be involved in DNA repair and replication for amino acid changes that are predicted to result in diminished function. Among aye-aye mothers, we found a nonsynonymous SNP in the gene *BLM* shared in heterozygous state by the two females that transmitted higher numbers of mutation, to the exclusion of all other parents. As was explained above, if this allele led to a hypermutator phenotype, the action of any such effect would have to be confined to advanced ages; it also would not explain the lack of a paternal age effect. Nevertheless, sampling additional pedigrees in which either older males or older females show variation in this gene would help to test its effects. We also observed multiple substitutions shared by all aye-ayes in *PRKDC*, finding that it is likely a non-functional gene in all the lemuriform primates. The main phenotypes of mutations in the *PRKDC* gene in humans are cancers and immune disorders (https://www.ncbi.nlm.nih.gov/gene/5591); however, recent work also suggests a direct role in mutation repair [45]. Other promising candidate substitutions were found in the genes *MBD4* and *MUTYH*. *MBD4* was the first gene in mammals with clear evidence for genetic variation that affected germline mutation in females [37]. Interestingly, that variant increased the mutation rate in female rhesus macaques but did not lead to an effect of maternal age on mutation, as was observed here for the aye-aye. Patterns of mutation are also affected by variation in the gene *MUTYH* in mice [38], and similar *MUTYH* mutations in humans have effects that are specific to the children of female carriers [39]. However, in both species the major effect of mutations in *MUTYH* is an increase in C > A mutations across the genome, a pattern observed in gray mouse lemurs [19] but not in aye-ayes (S2 Fig). It is also possible that the change in mutation patterns found in this study is due to an unmeasured environmental factor. Intriguingly, both the aye-aye and the gray mouse lemur studies were conducted on individuals housed in the DLC. Future work should strive to collect information on mutation rates from strepsirrhines kept in other facilities to test for the possible effect of shared environment.

One possibly contradictory finding is that long-term patterns of sex-biased mutation in all strepsirrhines are apparently male-biased (Fig 3). There are multiple possible factors that could explain the discrepancy between these results and the aye-aye pedigree mutation rates. First, female-biased mutation may be extremely rapidly evolving (or highly plastic) and may have recently shifted in aye-ayes. However, pedigree-based estimates of sex-biased mutation generally

match phylogeny-based estimates [2], so the aye-aye would have to show an unusual shift compared to other mammals. A second possible explanation is that the male-bias measured by the phylogenetic approach is driven largely by the sex bias present at the average time of reproduction in nature, averaged across millions of years. This age may have been quite different in the past—or even in the wild at present—compared to the captive families represented by our pedigrees. Little is known about the age of reproduction of aye-ayes in the wild, though it is well-established for mouse lemurs that reproductive maturity is achieved by one year of age in both the wild (e.g., [46]) and in captivity [23]. However, in captivity aye-ayes show a number of differences in female reproduction relative to other strepsirrhines. For instance, aye-ayes have much longer gestation periods than other lemuriforms, a much older age at first conception in females (3.66 years), and a much older age at last conception in females, despite having similar total lifespans relative to body size [23]. Researchers at the DLC have observed that all four of the original females in the aye-aye colony there have given birth after the age of 15, some of them multiple times. There are obviously many unique aspects to aye-aye female reproduction, and our results may therefore reveal a mechanism of female mutation bias unique to aye-ayes. Unfortunately, this female bias is not observable via comparative studies of substitution rates and is only accessible via much more expensive and time-consuming pedigree studies designed to include a wide range of maternal and paternal ages.

Male mutation bias was first uncovered by Haldane [47] and was confirmed by early studies of molecular evolution [48,49]. Although male bias was thought to be due to continuing replication in the male germline post-puberty (female germline replication is complete before birth), multiple lines of evidence have cast doubt on germline replication as the dominant factor in generating male bias [16,17]. However, a major impediment to uncovering the causes of male mutation bias is its universality, as all previous studies—at least when parents cover a broad enough age range—have found similar patterns. We currently know very little about germline replication or germline protection and stability in aye-ayes, especially as to whether these might differ between the sexes or from other mammals. Aye-ayes, and possibly other strepsirrhines, therefore represent an important new system that can be used to understand sex-biased mutation.

## Materials and methods

### Sample collection and DNA extraction

Aye-aye blood samples were collected using EDTA-coated tubes, and baboon blood samples were collected in PAXgene blood DNA collection tubes. High-quality genomic DNA was extracted from aye-aye samples using GenFind V3 kit (Beckman Coulter Life Sciences) and from baboon samples using PAXgene Blood DNA kit (Qiagen) following manufacturers' instructions. All blood samples were obtained from previously collected and banked materials, thus requiring no procedures involving live animals.

### Preparation of sequencing libraries

Whole genome sequencing data were generated for aye-aye and baboon samples at the Baylor College of Medicine Human Genome Sequencing Center using established methods. Libraries were prepared using KAPA Hyper PCR-free library reagents (KK8505, KAPA Biosystems) on Beckman robotic workstations (Biomek FX and FXp models). DNA (750 ng) was sheared into fragments of approximately 200–600 bp using the Covaris E220 system (96 well format, Covaris, Woburn, Massachusetts) followed by purification of the fragmented DNA using AMPure XP beads. A double size-selection step was used, with different ratios of AMPure XP beads, to select a narrow size band of sheared DNA molecules for library preparation. DNA end-repair and 3′-adenylation were performed in the same reaction followed by ligation of the Illumina

unique dual barcode adapters (Cat# 20022370) to create PCR-Free libraries. Each library was evaluated using the Fragment Analyzer (Advanced Analytical Technologies, Ames, Iowa) to assess library insert size and the presence of remaining adapter dimers. Finally, we carried out a qPCR assay with the KAPA Library Quantification Kit (KK4835) using their SYBR FAST qPCR Master Mix to estimate the size and quantification.

## Whole-genome DNA sequencing

Illumina whole-genome libraries were pooled and sequenced using S4 flow cells and the NovaSeq SBS reagent kit v1.5 on the NovaSeq 6000 according to manufacturer's instructions. A pool of sequencing libraries (about 250 pM) was loaded onto the S4 flow cell using the XP 4-lane kit. Cluster generation and 2 × 150 cycles of sequencing were performed to generate 150 bp paired-end reads with on-instrument base calling. The sequence data were transferred from the instruments for further processing, including mapping, using the HGSC workflow management system (Mercury v17.5; [50]).

## Read mapping and variant calling

BWA-MEM (v0.7.15; [51]) was used to map all sequences to the appropriate whole genome reference assembly, ASM2378347v1 (GCA_023783475.1) [52] and DauMad_v1_BIUU (GCA_004027145.1) for aye-ayes or Panu3.0 (GCA_000264685.2) for baboons [53]. The baboon data from Wu and colleagues [12] were also re-mapped and subject to all of the following steps. To identify multiple reads potentially originating from a single fragment of DNA and to mark them in the BAM files, we used Picard MarkDuplicates (v2.6.0; http://broadinstitute.github.io/picard/). Variants were then called using GATK (v4.2.2.0; [54]) following best practices, and a jointly called VCF file was generated. The hard filters suggested by the developers of GATK (https://gatk.broadinstitute.org/hc/en-us/articles/360035532412?id=11097) were applied to the SNVs and all variants failing filters were removed. We then used GATK VariantAnnotator to remove SNVs with an allelic balance for heterozygous calls (ABHet = ref/(ref + alt)) ABHet < 0.2 or ABHet > 0.8. All indels were removed. The average sequencing coverage used for SNV genotype calls in the samples was 39.1X for aye-ayes and 43.5X for baboons. Note that one individual from Wu and colleagues [12] showed very low read-depth after mapping (8.7X) and was removed from all downstream analyses (individual 1X3656).

## Identifying mutations

We used a standard set of methods for identifying autosomal DNMs, one that we have used previously multiple times [10,11,44] and that has been shown to have high accuracy [26]. An initial set of candidate mutations was identified as "mendelian violations" in each trio from the GATK results described above. We only considered violations where both parents are homozygous for the reference allele and the offspring was heterozygous for an alternate allele, in order to minimize error [55]. We then apply the following filters to the mendelian violations to get a set of high-confidence candidate DNMs:

1. Read-depth at the candidate site must have a minimum of 15 and a maximum of [ $\bar{X}$ + ($4\sqrt{\bar{X}}$ )] reads for every individual in the trio, where $\bar{X}$ represents the mean of the individual being considered [56].

2. High genotype quality (GQ) in all individuals (GQ > 60).

3. Candidate mutations must be present as a heterozygote from both GATK and bcftools [57]. (Candidates not called by bcftools are often due to remapping errors by the GATK HaplotypeCaller.)

4. Candidate mutation alleles must be present on reads from both the forward and reverse strand in the offspring.

5. Candidate mutation alleles must not be present in any reads from either parent.

6. Candidate mutation alleles must not be present in any other samples (except siblings).

7. Candidate mutation must have allelic balance >0.35 in the offspring.

The same criteria were used for both aye-ayes and baboons.

## Estimating the per-generation mutation rate

To estimate the mutation rate per-base per-generation, we divided the number of DNMs by the number of bases at which mutations could have been identified. As in previous work, we applied existing strategies that consider differences in coverage and filtering among sites [5,10,11], and that estimate false negative rates from this filtering. We estimate the total number of sites at which a mutation might be identified, the number of "callable sites," as a product of the number of sites across the genome that meet the sequencing depth filters as applied to DNMs, and the estimated probability that given such a site was a true DNM, that it would be identified correctly as such. The mutation rate is then calculated as:

$$\mu_i = \frac{N_{\text{mut},i}}{2 * \sum_x C_i(x)} \tag{1}$$

where $\mu_i$ is the per-base mutation rate for trio $i$, $N_{\text{mut},i}$ is the number of DNMs identified in trio $i$, and $C_i(x)$ is the callability, the probability that the genotype of a true DNM is correctly called, at site $x$ in that trio. Our approach assumes that the probability of correctly calling sites in each individual is independent, allowing us to estimate $C_i(x)$ as:

$$C_i(x) = C_c(x)C_p(x)C_m(x) \tag{2}$$

where $C_c$, $C_p$, and $C_m$ are the probability of calling the child, father, and mother genotypes correctly for trio $i$. We estimated these values by applying the same set of stringent mutation filters to high-confidence genotype calls from each trio. For heterozygous variants in the child, we estimated

$$C_c(x) = \frac{N_{\text{filtered}}}{N_{\text{all}}} \tag{3}$$

where $N_{\text{all}}$ is the number of variants in the offspring where one parent is homozygous reference and the other parent is homozygous alternate, giving us high confidence in the child heterozygote call, and $N_{\text{filtered}}$ is the set of such calls that pass our child-specific candidate mutation filters. The callability of genotypes in the parents, $C_p(x)$ and $C_m(x)$, were estimated in a similar manner, by calculating the proportion of high-confidence genotypes that pass the parent-specific filters for candidate mutations. This approach assumes no (or very few) false positives, an assumption corroborated by previous results applying these methods and the Sanger sequencing conducted here.

## Sanger sequencing

Candidate DNMs were validated by re-sequencing using PCR amplification and Sanger sequencing. PCR and sequencing primers were generated via manual primer design utilizing Primer3 [58]. The average amplicon length was 391 bp across loci. PCR amplifications were

performed using an Eppendorf Mastercycler ep 384 PCR System. Amplicons were sequenced using 3500XL Genetic Analyzer, Applied Biosystems. All bases with a Phred quality score of Q20 or greater within covered regions were analyzed using the Mutation Surveyor v5.2.0 (https://softgenetics.com/products/mutation-surveyor/), and genotypes were verified by manual observation of fluorescence peaks.

## Assigning mutations to a parent of origin

We used the software POOHA (https://github.com/besenbacher/POOHA; [4,5,29]) to assign parent of origin to the aye-aye and baboon DNMs identified in this study (in both the newly sequenced baboons and those sequenced in Wu and colleagues [12]). The rhesus macaque DNMs and parental assignments were taken from two previous studies and analyzed together [4,11], as were the assignments for humans [6]. POOHA uses read tracing (sometimes called "read-backed phasing" or "read-based phasing") to assign the parent of origin. Read tracing uses informative heterozygous SNPs found in the same read or paired-read as the DNM to assign the parent of origin. POOHA requires a VCF file for each trio and a BAM file only from the child. We provided data in a window 1,000 bp upstream and 1,000 bp downstream of each DNM site.

Only for the aye-aye DNMs, we also attempted to assign the parent of origin using the software Unfazed (https://github.com/jbelyeu/unfazed; [32]). Unfazed uses "extended" read tracing to find parent-of-origin informative heterozygous SNPs that may not be in the same read or read-pair as the DNM. Instead, SNPs that are informative about haplotypic phase in the child and that connect the DNM to parent-of-origin informative SNPs effectively "extend" the region that can be used. Again, we used data in a window 1,000 bp upstream and 1,000 bp downstream of each DNM site.

## Estimating effects of parental age

To estimate the effects of parental age on the mutation rate in each species, we used two approaches. Our first approach models the number of mutations assigned (phased) to each parent in a trio, controlling for the fraction of all mutations that were phased in that trio. We estimated the parental age effect using a Poisson regression with the number of phased mutations from each parent separately as response variables. That is, the number of maternally and paternally phased mutations were modeled as $N_M \sim \text{Poisson}(\mu_M)$ and $N_P \sim \text{Poisson}(\mu_P)$, and fit to a generalized linear model with an identity link function. Here $\mu_M$ and $\mu_P$ account for both the total number of sites and the total number of phased mutations in a trio $i$ as

$$\mu_i = \beta_0 + \beta_P \zeta_{P,i} \tag{4}$$

where $\beta_0$ and $\beta_P$ are the intercept and paternal coefficients, respectively, for paternally phased mutations. We also have

$$\zeta_{P,i} = L_i S_i X_{P,i} \tag{5}$$

where $L_i$ is the diploid callable genome size in trio $i$ (equivalent to the denominator in equation 1), $S_i$ is the total number of mutations identified in the trio, and $X_{P,i}$ is the paternal age of parents for the same trio. The regression for maternally phased mutation used the same formula with separate intercept and maternal coefficients ($\beta_0$ and $\beta_M$). In practice, we added a pseudocount of 1 to the number of maternally and paternally phased mutations, respectively, when performing this regression in order to avoid non-positive values while using an identity link function.

Our second approach considered the total number of mutations, $N$, in each trio with a Poisson regression, $N \sim \text{Poisson}(\mu)$. These mutations do not have to be assigned to a parent of origin. We applied an identity link function and modeled the per site mutation rate for a given trio $i$ as:

$$\mu_i = \beta_0 + \beta_P Z_{P,i} + \beta_M Z_{M,i} \tag{6}$$

where $\beta_0, \beta_P, \beta_M$ are the intercept, paternal, and maternal coefficients, respectively. And

$$Z_{P,i} = L_i X_{P,i} \quad Z_{M,i} = L_i X_{M,i} \tag{7}$$

where $X_{P,i}$ and $X_{M,i}$ are the paternal and maternal age of parents, respectively, for trio $i$, and $L_i$ is the diploid callable genome size from the same trio.

In Fig 2, we plotted the inferred the number of mutations each parent transmitted by dividing the number of mutations phased to a given parent by the fraction of mutations phased in that trio (e.g., inferred paternal count = paternally phased mutations/(paternally phased + maternally phased)). These inferred numbers are not used in any regression.

## Estimating sex-biased substitution from comparative data

We used the 241-way mammalian alignment from the Zoonomia Consortium (v2; [59]), which offers the advantage of allowing any species of interest to be used as a reference. We converted the hierarchial alignment format (HAL) file to multiple alignment format (MAF) using the hal2maf tool (https://github.com/ComparativeGenomicsToolkit/hal/; [60]), selecting *M. murinus* (gray mouse lemur) as the reference genome. During this initial step, we extracted a subset of all strepsirrhine species present in the original alignment of 241 species, each of which was then assessed for the number of aligned sites. Based on this assessment, we selected 10 species with sufficient high-quality alignments for further analysis. To ensure that our estimates of substitution rates reflected mostly mutational processes, we focused on non-coding regions that are orthologous across all these species. Additionally, we excluded regions in the reference genome that were not assembled to the chromosome level, as well as their homologous regions in the other species, to avoid potential biases in our comparison between the X chromosome and autosomes. Likewise, sequences overlapping with pseudo-autosomal regions on the X chromosome were discarded since they behave like autosomes due to their homology with the Y chromosome. This filtering was performed using the maf_parse tool in PHAST (http://compgen.cshl.edu/phast/; [61,62]), using BED files annotated in the reference genome.

Branch lengths (a proxy for substitution rate) for the X chromosome and autosomes were estimated using the phyloFit program in PHAST (http://compgen.cshl.edu/phast/; [61]), with the tree topology from Zoonomia. The analysis employed an unrestricted single nucleotide model (UNREST) using the expectation-maximization algorithm. We executed six independent runs of phyloFit, each with random parameter initialization, and selected the replicate with the highest likelihood to ensure robustness. These analyses were carried out separately for the X chromosome and each autosome, with the average of all autosomes used as the final value.

To infer the male-to-female substitution rate ratio ($\alpha$), we used the equation [34]:

$$\alpha = \frac{4 - \left(\dfrac{3X}{A}\right)}{\left(\dfrac{3X}{A}\right) - 2}, \tag{8}$$

where *X* represents the length of a branch on the X chromosome and *A* represents the length of the same branch for the autosomes. This equation assumes that the underlying mutation rate is the same on the X and autosomes within a single sex, so that any difference in long-term substitutions is driven by the amount of time the X spends in males relative to females. Male-biased mutation then results in values of $\alpha > 1$, while female-biased mutation results in values of $\alpha < 1$.

To calculate confidence intervals on $\alpha$ for the branch leading to aye-ayes, we calculated separate branch lengths for each of 31 autosomes (one chromosome was excluded for having too few aligned bases). An $\alpha$ value was then calculated for each chromosome, and a 95% confidence interval around the mean was estimated using these values.

### Candidate DNA repair genes for female mutation bias

To investigate whether differences in DNA repair mechanisms contribute to the observed shift in the sex-specific mutation bias in aye-ayes, we focused on a set of 219 genes known to be involved in DNA repair (https://www.mdanderson.org/documents/Labs/Wood-Laboratory/human-dna-repair-genes.html; S4 Table). This list represents an inventory of human DNA repair genes initially published by Wood and colleagues [63] and subsequently expanded by incorporating additional work (e.g., [64]). These genes are associated with critical DNA repair processes, including those linked to genetic instability and sensitivity to DNA-damaging agents.

### Identifying variants in DNA repair genes

We looked for variants in the 219 candidate genes in two datasets. To find aye-aye and lemuriform-specific substitutions, we extracted 239 primate species from the 447-way whole-genome multiple sequence alignments in Kuderna and colleagues [65]. Alignment data from aye-aye, 30 lemur species, and 7 sifaka species were downloaded as MAF files. The software MafFilter [66] was used to extract sub-alignments of the 219 candidate genes based on human GENCODE v46 annotations and human genome coordinates from the MAF files. To find allelic variants unique to the two aye-aye mothers that gave birth at the oldest ages (100943 and 100949), we examined all SNPs in the 219 genes using coordinates based on the human GENCODE v46 annotations.

We generated variant call format (VCF) files for any variants found in the primate species included in the MAF alignments using MafFilter, as well as a VCF files for SNPs specific to the two focal mothers. The resulting VCF files were annotated using the Ensembl Variant Effect Predictor v108 [67] and CADD v1.6 [35], which predict the potential impact of variants on gene function. The annotated VCF files were used to identify substitutions seen only in aye-aye, substitutions in aye-aye and any other of the 37 lemuriforms compared to all other primates, or SNPs specific to the two focal mothers.

### Supporting information

**S1 Fig. Identifying X-linked contigs and confirming the sex of each aye-aye individual.** For each of the 6,289 contigs greater than 100 kilobases in length, the read-depth and heterozygosity in each individual was measured. Each contig is shown here represented by two points: one for the male mean and one for the female mean. Colored points represent contigs that have significantly ($P < 0.05$) different read-depth OR heterozygosity between the sexes AND that have <25X read-depth in males; these were inferred to be X-linked and were not used to identify mutations. Contigs with significant differences in read-depth or heterozygosity but

that did not have read-depth <25X in males are in darker gray ("unfiltered"). No individuals were observed whose assigned sex was discordant with their expected read-depth and heterozygosity. The data underlying this figure can be found in S1 Data.
(JPEG)

**S2 Fig. Mutation spectrum in aye-ayes.** (**A**) The frequency of each mutation type is shown, among all 647 DNMs identified. (Mutation types represent their reverse-complement as well.) Mutations at CpG sites accounted for 17.7% of all mutations. (**B**) The frequency of all mutations assigned as coming from male parents. (**C**) The frequency of all mutations assigned as coming from the two female parents at the three oldest ages of birth. (**D**) The frequency of all mutations assigned as coming from all other female parents and births. The data underlying this figure can be found in S1 Data.
(JPEG)

**S3 Fig. Relationship between parental age and number of transmitted mutations in aye-ayes using alternative methods.** (**A**) This figure contains the same data as are represented in Fig 2A, again showing mutations assigned to parents using the software package POOHA. (**B**) This figure contains the same data as are represented in Fig 2A, but observed mutation counts are from assignments to parents using the software package Unfazed. Significance of coefficients for age effect in Poisson regression: Paternal $P = 0.24$, Maternal $P = 0.04$. The data underlying this figure can be found in S1 Data.
(JPEG)

**S4 Fig. Total number of mutations in each aye-aye offspring as a function of (A) maternal age or (B) paternal age.** The data underlying this figure can be found in S1 Data.
(JPEG)

**S5 Fig. An example of a mutation in aye-ayes confirmed by Sanger sequencing.** Electropherograms for the region around an inferred de novo mutation (indicated by an arrow in the proband) are shown for the family in which the mutation appeared. Only the proband is heterozygous at this position. ID numbers refer to Fig 1A.
(JPEG)

**S6 Fig. Relatedness among all parents in the first generation ("founders").** We estimated the coefficient of relatedness (CoR) between all pairs of founders based on their genotypes at a random sample of 100,000 variable sites. The CoR of the two mothers in our sample that had children at the oldest ages is shown as a red dot. Our estimate of the CoR follows the formula from Pedersen and Quinlan [69], in which a CoR of 0 or less indicates unrelated individuals. The data underlying this figure can be found in S1 Data.
(JPEG)

**S7 Fig. Allelic balance of candidate and final mutations in aye-ayes.** Distribution of allelic balance for candidate mutations is shown, highlighting the set of DNMs accepted after applying the threshold of 0.35. All candidates less than this threshold were discarded (gray), while all candidates greater than this threshold were kept as the final set (pink and red). The red bars indicate the allelic balance of all maternally transmitted mutations for the three oldest ages of maternal conception. The data underlying this figure can be found in S1 Data.
(JPEG)

**S8 Fig. Relationship between parental age and number of transmitted baboon mutations.** (**A**) Data from the same trios analyzed and used in Wu and colleagues [12]. (**B**) Data from all trios sequenced in Wu and colleagues [12]. The datapoints that have been added relative to

panel A are highlighted in red. (**C**) All data used here, including all datapoints from panel B plus trios sequenced for the first time in this study (highlighted in green). The data underlying this figure can be found in S1 Data.
(JPEG)

**S1 Data.   Data underlying all figures.**
(XLSX)

**S1 Table.   Mutations in aye-ayes.** The chromosome, position, and parent-of-origin for each of the mutations identified in aye-ayes.
(CSV)

**S2 Table.   Heterozygosity in aye-ayes.** Statistics on single-nucleotide variants from aye-ayes in our sample.
(DOCX)

**S3 Table.   Mutations in baboons.** The chromosome, position, and parent-of-origin for each of the mutations identified in baboons.
(CSV)

**S4 Table.   Candidate DNA replication and repair genes.** List of genes that were analyzed for their possible mechanistic role in the aye-aye mutation bias.
(CSV)

**S5 Table.   Polymorphisms in candidate genes among aye-aye mothers.** List of polymorphisms found in mothers that transmitted a "high" number of mutations compared to other mothers, and all other parents.
(CSV)

**S6 Table.   Deleterious candidates unique to aye-ayes.** List of all predicted deleterious substitutions in candidate genes unique to the aye-aye.
(CSV)

**S7 Table.   Deleterious candidates unique to lemuriforms.** List of all predicted deleterious substitutions in candidate genes unique to lemuriforms.
(CSV)

## Acknowledgments

We thank Donna Muzny, Harsha Doddapaneni, Richard Gibbs and all the members of the sequence production teams at the Baylor College of Medicine Human Genome Sequencing Center for their efforts and expertise. We also thank the staff at the Duke Lemur Center and the Keeling Center for Comparative Medicine and Research for their efforts. This is DLC publication number 1608.

## Author contributions

**Conceptualization:** Richard J. Wang, Jeffrey Rogers, Matthew W. Hahn.

**Data curation:** Yadira Peña-García, Muthuswamy Raveendran, R. Alan Harris, Thuy-Trang Nguyen.

**Formal analysis:** Richard J. Wang, Yadira Peña-García, R. Alan Harris.

**Funding acquisition:** Richard J. Wang, Jeffrey Rogers, Matthew W. Hahn.

**Investigation:** Richard J. Wang, Yadira Peña-García, Muthuswamy Raveendran, R. Alan Harris, Thuy-Trang Nguyen, Matthew W. Hahn.

**Methodology:** Richard J. Wang, Matthew W. Hahn.

**Project administration:** Matthew W. Hahn.

**Resources:** Anne D. Yoder, Joe H. Simmons.

**Software:** Richard J. Wang, Yadira Peña-García.

**Supervision:** Marie-Claude Gingras, Jeffrey Rogers, Matthew W. Hahn.

**Validation:** Muthuswamy Raveendran, Marie-Claude Gingras, Yifan Wu, Lesette Perez.

**Writing – original draft:** Matthew W. Hahn.

**Writing – review & editing:** Richard J. Wang, Yadira Peña-García, Muthuswamy Raveendran, R. Alan Harris, Anne D. Yoder, Jeffrey Rogers, Matthew W. Hahn.

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
