## [Editor Report · Decision Letter 0]

10 Sep 2024

Dear Dr Hahn, 

Thank you for submitting your manuscript entitled "Unprecedented female mutation bias in aye-ayes" for consideration as a Discovery Report by PLOS Biology. Please note that I am currently handling your manuscript whilst my colleague Roland Roberts is away from the office this week. 

Your manuscript has now been evaluated by the PLOS Biology editorial staff, as well as by an academic editor with relevant expertise, and I am writing to let you know that we would like to send your submission out for external peer review.

Once your full submission is complete, your paper will undergo a series of checks in preparation for peer review. After your manuscript has passed the checks it will be sent out for review. To provide the metadata for your submission, please Login to Editorial Manager (https://www.editorialmanager.com/pbiology) within two working days, i.e. by Sep 12 2024 11:59PM.

Kind regards,

Richard 

Richard Hodge, PhD

rhodge@plos.org

On behalf of:

Roland Roberts, PhD

rroberts@plos.org

---

## [Decision Letter · Decision Letter 1]

25 Oct 2024

Dear Matt,

Thank you for your patience while your manuscript "Unprecedented female mutation bias in aye-ayes" was peer-reviewed at PLOS Biology. It has now been evaluated by the PLOS Biology editors, an Academic Editor with relevant expertise, and by three independent reviewers. 

You'll see that reviewer #1 calls the findings “striking” and “intriguing,” but notes that only 3 offspring have a maternal bias and asks whether maternal age should be factored in. Reviewer #2 is also very positive, and notes the same n=3 issue, asking you to flag this more prominently and to rule out a shared mutator phenotype. S/he also asks you to analyse mutational spectra, plus a few other analyses and clarificatory questions. Reviewer #3 asks explicitly about the apparent disparity with de Manuel et al., wants some discussion of mechanism, requests several further analyses, additional sequence validation, and (like rev #2) asks about mutational spectra.

In light of the reviews, which you will find at the end of this email, we would like to invite you to revise the work to thoroughly address the reviewers' reports.

Given the extent of revision needed, we cannot make a decision about publication until we have seen the revised manuscript and your response to the reviewers' comments. Your revised manuscript is likely to be sent for further evaluation by all or a subset of the reviewers.

**IMPORTANT - SUBMITTING YOUR REVISION**

*Re-submission Checklist*

*Published Peer Review*

*PLOS Data Policy*

*Blot and Gel Data Policy*

Sincerely,

Roli

Roland Roberts, PhD

Senior Editor

PLOS Biology

rroberts@plos.org

REVIEWERS' COMMENTS:

Reviewer #1: 

Wang et al reveal a striking new result in the field of germline mutation rates and sex bias in mammalian genomes by finding evidence for female mutation bias in aye-ayes.

They use whole-genome sequencing in multiple pedigrees and employing standard methods to call and phase mutations. As a control, they use WGS on baboon pedigrees.

This is a very intriguing result, and I would ask the authors to address a few questions and concerns.

1. Looking closely at Figure 1A and Table 1, only three aye-aye offspring have a maternal bias in mutation bias. They are 100947 with 7 paternal, 23 maternal; 100933 with 16 paternal and 23 maternal; 100945 with 11 paternal and 16 maternal. They other 9 offspring in the study have a paternal bias and that bias is often quite strong (9:1 in 100938, 11:1 in 100941, 9:3 in 100950, 11:0 in 100942, and 9:1 in 100939). Thus alpha is consistent with a paternal bias in the majority of the offspring. What do the authors make of this? Moreover, for two of the 3 offspring with a maternal bias, they are in a family where two other offspring show a slight paternal bias. To me, it looks like maternal bias really only takes effect when mothers are older and perhaps also older than the father, which is the case for two of the three offspring with a maternal bias. This advanced maternal age does not seem to be the case for the baboons and it is rarely the case in human. I would ask that the authors consider this as a confounder.

2. Since the effect is only seen in 1/2 of the offspring for one family, none of the offspring for another family, and 1/1 offspring in the third family, is it possible that sample swaps happened for subset?

3. Could genome-wide differences in depth between mother and father genomes bias the assignment of DNMs to one sex versus the other? That is, allelic dropout in the father?

Reviewer #2:

This paper presents a striking example of female-biased mutation, which is more-or-less accurately described as an unprecedented finding. This finding could be viewed as preliminary due to the study's small sample size (just two females and three offspring drive the result). However, publication is justified in my view since 1) larger studies of nonhuman primate mutagenesis are rare, and 2) the paper's data provide important information about where future studies of primate mutagenesis might want to focus their efforts and resources. Below, I suggest a few small analyses that might shed light on the nature of this maternal age effect, and I also have a few suggestions for further improving the clarity and accuracy of the text.

Figures 1 and 2 make it fairly transparent that the finding of female mutation bias is driven by just three trio datapoints, two of which share the same mother. However, I didn't see this fact explicitly mentioned anywhere in the paper text, and it seems appropriate to discuss this along with the other discussion section caveats. It is striking that these are the only three trios where the mother is over the age of 15 years, and that gives me confidence that the result has a good chance of being a universal feature of aye-aye biology, but it's also possible that these two mothers are unusual in some way. They might have a mutator phenotype like the one described in the Stendahl, et al paper (though one that is age-dependent, unlike that phenotype) or they might be unusual in their ability to reproduce at an advanced age. The discussion section mentions that not much is known regarding aye-aye generation times in the wild, but is there any information regarding the proportion of female aye-ayes who have successfully given birth to live young at ages of 15+ years within the Duke lemur facility? It would also be useful to clarify whether these mothers have a known genealogical relationship to each other, or whether they share an unusual amount of their genomes identical by descent.

The analysis of aye-aye DNA repair gene substitutions was interesting, but it could be improved by the addition of some more context. Do the baboons and macaques have similar numbers of apparently deleterious DNA repair gene substitutions? Do the results change at all if we look at all DNA repair gene mutations that are common to the two mothers who had children at ages of 15+ years, rather than just DNA repair gene mutations that appear fixed in aye-ayes?

It would be straightforward and informative to look at whether the aye-aye maternal age effect has similar spectral characteristics to the maternal age effect observed in humans. For example, are the three advanced maternal age trios enriched for C>G mutations relative to the other aye-aye trios, as has been seen for human advanced maternal age? If so, is there any evidence for clustering of the additional C>G mutations into genomic hotspot regions that might also show elevated levels of C>G substitutions or polymorphisms? Does the dependence of mutation rate on maternal age fit an exponential curve significantly better than a linear regression line, as has been seen in some human studies but not others? 

I actually did not share the authors' worry that the X:autosome substitution results appear to contradict the DNM results. It seems unlikely that reproduction at 15+ years is any more common in nature than it is in captivity, given that life expectancy is probably longer in captivity and nutrition is likely more consistent. If one were to average together the maternal and paternal mutation loads across all the trios in the dataset, it looks likely to be male biased. I think it could be informative to do that calculation explicitly and test whether it yields an expected alpha value that is similar to the one computed phylogenetically. It would also be useful to mention whether the de novo mutation rate and spectrum appears to be any different on the aye-aye X chromosome compared to the autosomes, given that such a spectral difference was previously detected in humans by Agarwal & Przeworski. Is there any power to detect whether the maternal bias found on the autosomes also applies to the X?

Minor comments:

I assume that the authors tried and failed to find an explanation within the Wu, et al. paper for why these additional trios were not included in their analysis, but an explanation would be worthwhile to include if possible. It would also be useful to explicitly test whether the distribution of paternal ages is responsible for the differences between the two papers' results, as presented as a conjecture in lines 164-65. I also noticed that the two papers find quite different maternal age effects—in the current paper, the effect is negative, while in the Wu, et al. paper, it is larger than the paternal age effect. Do age distributions also appear to explain this contradiction?

I was surprised to see that the paternal and maternal regression p values in line 186 were so different, given that the maternal and paternal ages in Table 1 appear correlated to my eyes. Given this correlation, I would have expected the highly significant maternal age effect on the total mutation load to create at least some appearance of a paternal age effect. Any idea why this is not the case?

Lines 353-54 state that the Stendahl, et al. macaque paper provided the first "clear evidence for genetic variation that only affected germline mutation in females." I don't think Stendahl et al. showed this—they demonstrated that their MBD4 allele had a maternal effect, but they didn't have the right data to test whether or not it also had a paternal effect.

The reason I say above that I "more-or-less" agree with the use of the term "unprecedented" is that the Stendahl paper does contain examples of trios where the maternal lineage is the origin of the majority of the mutation load. Also, the Wu, et al. paper finds a maternal age effect that is nominally larger than the paternal age effect. The manuscript even says in line 326 that their results "have some precedent." However, these examples are minor enough that I would say the title is justified.

Reviewer #3:

This study presents an interesting and unexpected discovery regarding the patterns of de novo mutation transmission in aye-ayes primates. In contrast to the well-established male mutation bias observed across mammals, where males typically transmit more mutations to their offspring than females, the authors find that in aye-ayes, the opposite occurs. Maternal age has a much stronger effect on mutation rates than paternal age, and older mothers transmit significantly more mutations than older fathers, which is unprecedented in primates.

The authors utilized whole-genome sequencing data from 12 aye-aye trios and 4 baboon trios, using the same computational and experimental approaches for consistency. While baboons, rhesus macaques, and humans showed the expected male-biased mutation rates, the aye-aye data revealed a female mutation bias, particularly with increasing maternal age. This result suggests that mutation bias is not a fixed trait in mammals and can evolve in closely related species. This work challenges long-held assumptions about sex-biased mutation rates in mammals and opens the door to further investigations in this space.

Overall, the finding is certainly novel, and if it holds true, it will be highly significant for the field and of great interest to the readers of the journal. However, given that sex bias in germline mutation is a well-studied subject, with consistent patterns reported across taxa, this study would benefit from further follow-up validations to strengthen the robustness of the results. These additional efforts would help ensure that the observed maternal mutation bias is representative of the aye-aye species, rather than a peculiarity of the sampled individuals. Below, I have listed some of my concerns that I hope will help improve the clarity and overall robustness of the findings.

1- The trio data from aye-ayes was previously sequenced by de Manuel et al. (2022), and they reported a male-biased mutation pattern like that observed in other primates and most mammals. Given this, I'm unclear why these new results would suggest such a strikingly different outcome, especially with the strong maternal bias and pronounced age effect. If this is accurate, there must be some fundamental biological differences causing this maternal bias, which is quite unusual, particularly given that oogenesis in mammals does not involve post-birth cell divisions. While recent work in humans (Abascal et al., 2021; Moore et al., 2021) suggests that mutations can accumulate with age in post-mitotic tissues, the rate of mutation accumulation in the maternal germline here seems unexpectedly high. I am not sure if this is discussed in depth in the current manuscript - especially since the outcome from the phylogeny trees are quite opposite to their findings! 

2- I am unclear whether any of the 12 trios share maternal or paternal line. A clear analysis of relatedness within these trios is important for strengthening the significance of the findings. I suggest comparing SNPs between all 12 trios to clarify any genetic relationships by genotype.

3- Additionally, it would be helpful to provide more clarity on how the overall mutation burden in these 12 trios, after accounting for parental age effect, compares with similar primates or human. This comparison would give more context to the findings.

4- Although paternal bias is consistently reported across taxa, there have been cases of maternal bias, as cited by the authors. In those instances, the maternal line often carries a hypermutator phenotype, meaning that the mother has an elevated germline or somatic mutation burden due to genetic factors such as MBD4 or MUTYH mutations, which impair DNA replication or repair. The analysis here does not seem sufficient to exclude the possibility of a hypermutator effect, particularly given the suspicion that some of these trios might be closely or distantly related. With the ~30-40X whole-genome sequencing, it's likely insufficient to identify enough coding mutations, particularly if these mutations are mosaic. Perhaps high depth targeted or exon sequencing of maternal genomes would add more clarity that none of these mums have a hypermutator germline. 

5- Given the importance of the findings, I would strongly recommend validating these de novo mutations (DNMs) using deep targeted sequencing across all trios. This would add much-needed robustness to the conclusions. Although the Sanger sequencing is useful approach to exclude possibility of germline mutation, it is fairly shallow and won't be able to detect any possibility of mosaic mutations. 

6- Additionally, it would be useful to expand this work on other trios from other centres to ensure this finding can be generalised and exclude possibility of certain. Environmental exposures that may cause hypermutation in these female lines. 

7- The observation of a negative maternal age correlation in baboons is unexpected. Although the authors achieved better phasing results in baboons, they did not seem to replicate the expected parental age effects seen in previous studies. While modest sample size might contribute to this, previous publications have not reported this negative correlation, raising concerns about potential confounders.

8- Is there any bias in the depth of coverage between maternal and paternal samples? The figure shows variation in mean coverage, but I don't see whether this was accounted for in the regression analyses. This could potentially affect the results, particularly for phased mutations.

9- There also seems to be variation in heterozygosity levels between maternal and paternal samples. This raises the question of whether some of the observed maternal bias might be influenced by inbreeding, where mothers contribute more to the progeny's genetic makeup.

10- It would be useful to explore whether there are differences in the mutation spectra between maternal and paternal lineages, as similar analyses have been conducted in previous studies (e.g., Rahbari et al. 2015 or Moore et al., 2021). Understanding if there are distinct mutational signatures would add depth to the findings.

11- Lastly, what are the variant allele fractions of the DNMs? How many of these are likely post-zygotic mutations? It would be particularly interesting to identify post-zygotic mutations in related trios by looking for shared DNMs between siblings. This could provide insight into whether some of the observed mutations arise post-fertilization.

Minor comment: 

Main figures can be improved and benefit from more panel with important information on overall burden between these trios, and other points that are mentioned above.

---

## [Decision Letter · Decision Letter 2]

19 Dec 2024

Dear Matt,

Thank you for your patience while we considered your revised manuscript "Unprecedented female mutation bias in aye-ayes" for publication as a Discovery Report at PLOS Biology. This revised version of your manuscript has been evaluated by the PLOS Biology editors, the Academic Editor and two of the original reviewers.

Based on the reviews and on our Academic Editor's assessment of your revision, we are likely to accept this manuscript for publication, provided you satisfactorily address the remaining points raised by the reviewers and the Academic Editor. Please also make sure to address the following data and other policy-related requests.

IMPORTANT - please attend to the following:

a) Please could you change the Title slightly to "Unprecedented female mutation bias in aye-aye lemurs," for those who don't know what an aye-aye is?

b) Please attend to the remaining requests from reviewer #2 AND from the Academic Editor (which can be found below the reviews). I should also say that reviewer #3, who was not able to return a review in time, was also broadly satisfied with your revisions.

c) I note that you say that no approval was needed, but you also say that blood samples were obtained from aye-ayes and baboons, so please could you clarify what the ethical situation is here (e.g. were the samples taken for veterinary purposes, or is Duke Lemur Center authorised to extract blood for research purposes?).

d) Please address my Data Policy requests below; specifically, we need you to supply the numerical values underlying Figs Figs 2ABCD, 3, S1, S2ABCD, S3AB, S4AB, S6, S7, S8ABC, either as a supplementary data file or as a permanent DOI’d deposition.

e) Please cite the location of the data clearly in all relevant main and supplementary Figure legends, e.g. “The data underlying this Figure can be found in S1 Data” or “The data underlying this Figure can be found in https://zenodo.org/records/XXXXXXXX

f) I note that you mention the reviewers ("Three reviewers also provided very constructive comments") in the Acknowledgements. While we appreciate the sentiment, this is against PLOS policy, so please could you remove this?

g) Please make any custom code available, either as a supplementary file or as part of your data deposition.

We expect to receive your revised manuscript within two weeks. 

*Published Peer Review History*

*Press*

Best wishes and Happy Holidays,

Roli

Roland Roberts, PhD

Senior Editor

rroberts@plos.org

PLOS Biology

ETHICS STATEMENT:

-- Please include the full name of the IACUC/ethics committee that reviewed and approved the animal care and use protocol/permit/project license. Please also include an approval number.

-- Please include the specific national or international regulations/guidelines to which your animal care and use protocol adhered. Please note that institutional or accreditation organization guidelines (such as AAALAC) do not meet this requirement.

DATA POLICY:

Regardless of the method selected, please ensure that you provide the individual numerical values that underlie the summary data displayed in the following figure panels as they are essential for readers to assess your analysis and to reproduce it: Figs 2ABCD, 3, S1, S2ABCD, S3AB, S4AB, S6, S7, S8ABC. NOTE: the numerical data provided should include all replicates AND the way in which the plotted mean and errors were derived (it should not present only the mean/average values).

CODE POLICY

DATA NOT SHOWN?

REVIEWERS' COMMENTS:

Reviewer #1:

Thank you for the revisions and the response to my concerns. Nice work!

Reviewer #2:

The authors have done a nice job with revisions. I only have a couple of minor remaining comments.

The new result on reconciling the phylogenetic estimate of alpha with the de novo mutation data appears to be mentioned for the first time in the discussion. As this is a new analysis, it seems like it should go in the results section about the phylogenetic alpha estimate. I also think this 8.97 years point estimate could benefit from some additional context. Since 8.97 years is earlier than most of the maternal ages in this dataset, I was initially unsure whether an average maternal age this young would be realistic. But when I googled the age of aye-aye puberty, I found that sexual maturity occurs between 2 and 3 years of age, which makes an average reproductive age of 8.97 years seem more reasonable than I initially thought given the age distribution in Figure 2. In addition to providing this biological context, it could be worth putting some kind of rough confidence interval around the 8.97 number. For example, given the lack of a paternal age effect, I infer that any distribution of paternal ages would be compatible with the alpha data; it is only the maternal age distribution that is constrained. It also seems important to note that this 8.97 number comes from a linear interpolation of the maternal age data, which actually looks more bimodal than linear, with basically no maternal age effect if you look at ages <15 years. So it seems unclear whether the maternal mutation load increases linearly versus remains low until age ~15 and then shoots up nonlinearly. Under the second model, the constraint on maternal age would be likely weaker than under a linear model. I don't expect the authors to discuss such possibilities exhaustively, but I would suggest giving some kind of maternal age distribution rather than just a point estimate, and discussing the assumptions and uncertainties associated with the calculation.

My other minor comment is that the paper's qualitative descriptions of regression strength do not seem consistent. Lines 194-195 talk about a "strong association" with P-value 0.03, while lines 283-284 talk about a "weak association" with P-value 0.04. It might be more appropriate to just describe both associations as "significant" or in some other way that reflects their similar P-values.

COMMENTS FROM THE ACADEMIC EDITOR:

The authors have addressed concerns well but I think they dismiss some arguments too easily and don't fully consider the mutational spectrum. They state in the response that a female bias isn't known in other species and cite humans. But it is known for C->T mutations (Goldmann et al. 2016; Jónsson et al. 2017), as Versoza et al note in the species considered here. Goldman et al for example say:

"...whereas maternally derived DNMs contain more C–T mutations (Bonferroni-corrected P = 2.7 × 10−3, Fig. 3a, Supplementary Table 24)."

They also say:

"Of note, we find an enrichment of maternal DNMs with motifs of APOBEC-mediated mutagenesis (χ2 test for enrichment P = 0.029), which are known to result from aberrant DNA double-strand break repair16. The efficiency of oocyte double-strand break repair is known to decrease in aging women16, which might result in a higher susceptibility to APOBEC-mediated mutations. "

I notice too a candidate gene is associated with DSB repair.

That signature is non-CpG mutations - the ones the current authors show are in excess in old females (suppl fig 2). The motif is TpCpN (from the above ref 16).

So a) female bias for some mutations is known, the one they see as being common in older females and b) they should check to see if the female excess is associated with APOBEC TpCN mutations. This all needs to be discussed and related to the prior literature.

Please also update all papers (Young et al is now published)

Goldmann JM, Wong WS, Pinelli M, Farrah T, Bodian D, Stittrich AB, Glusman G, Vissers LE, Hoischen A, Roach JC, et al. 2016. Parent-of-origin-specific signatures of de novo mutations. Nat Genet. 48(8): 935–939.

Jónsson H, Sulem P, Kehr B, Kristmundsdottir S, Zink F, Hjartarson E, Hardarson MT, Hjorleifsson KE, Eggertsson HP, Gudjonsson SA, et al. 2017. Parental influence on human germline de novo mutations in 1,548 trios from Iceland. Nature. 549(7673): 519– 522.

Versoza CJ, Ehmke EE, Jensen JD, Pfeifer SP. Characterizing the rates and patterns of de novo germline mutations in the aye-aye (Daubentonia madagascariensis). bioRxiv [Preprint]. 2024 Nov 11:2024.11.08.622690. doi: 10.1101/2024.11.08.622690. PMID: 39605388; PMCID: PMC11601268.

---

## [Editor Report · Decision Letter 3]

13 Jan 2025

Dear Matt,

Happy New Year! Thank you for the submission of your revised Discovery Report "Unprecedented female mutation bias in the aye-aye, a highly unusual lemur from Madagascar" for publication in PLOS Biology. On behalf of my colleagues and the Academic Editor, Laurence Hurst, I'm pleased to say that we can in principle accept your manuscript for publication, provided you address any remaining formatting and reporting issues. These will be detailed in an email you should receive within 2-3 business days from our colleagues in the journal operations team; no action is required from you until then. Please note that we will not be able to formally accept your manuscript and schedule it for publication until you have completed any requested changes.

Sincerely, 

Roli

Senior Editor

PLOS Biology

rroberts@plos.org